# Serological Changes in Anti-*Aspergillus* IgG Antibody and Development of Chronic Pulmonary Aspergillosis in Patients Treated for Pulmonary Tuberculosis

**DOI:** 10.3390/jof8020130

**Published:** 2022-01-28

**Authors:** Changwhan Kim, Jin-Wook Moon, Yong-Bum Park, Yousang Ko

**Affiliations:** 1Department of Internal Medicine, School of Medicine, Jeju National University Hospital, Jeju National University, Jeju 63241, Korea; masque70@naver.com; 2Division of Pulmonary, Allergy and Critical Care Medicine, Department of Internal Medicine, Kangdong Sacred Heart Hospital, Hallym University College of Medicine, Seoul 05355, Korea; cozybar@hanmail.net (J.-W.M.); bfspark2@gmail.com (Y.-B.P.)

**Keywords:** pulmonary tuberculosis, chronic pulmonary aspergillosis, anti-*Aspergillus* IgG antibody, seroconversion, serological change

## Abstract

Chronic pulmonary aspergillosis (CPA) is an important infection to understand in survivors of pulmonary tuberculosis (PTB). However, limited data are available regarding CPA development and its predisposing factors following PTB. We investigated the development of, and the predisposing factors for, CPA following the completion of PTB treatment. A total of 345 patients, with newly diagnosed culture-positive PTB (between January 2015 and December 2018), were included. Enrolled cases were categorized into four groups (persistently seronegative, seroconversion, seroreversion, and persistently seropositive) according to serological changes in their anti-*Aspergillus* IgG antibodies before and after PTB treatment. The patients were followed up for a median of 25.8 months. Ten (10/345, 2.9%) patients developed CPA at a median of 13.5 months after treatment completion, including seven (7/24, 29.2%) and three (3/73, 4.1%) in the seroconversion and persistently seropositive groups, respectively. Upon multivariate analysis, seroconversion of anti-*Aspergillus* IgG antibody (adjusted hazard ratio [HR], 25.21; 95% confidence interval [CI], 6.11–103.99; *p* < 0.001) and diabetic status (adjusted HR, 7.54; 95% CI, 1.93–29.50; *p* = 0.004) were independently associated with CPA development. The development of CPA in patients with PTB was observed in 2.9% of patients during post-treatment follow-up, and this was significantly associated with both the seroconversion of anti-Aspergillus IgG antibody and diabetes characteristics.

## 1. Introduction

Tuberculosis (TB) remains a major cause of morbidity and mortality worldwide. In 2018, 10.4 million people developed pulmonary tuberculosis (PTB), and 1.3 million people died from the disease [1]. Fortunately, the incidences and mortalities of TB infections are gradually decreasing due to global efforts. Consequently, survivors of PTB have become more common. Unfortunately, survivors of PTB may experience post-tuberculosis lung disease (PTLD). TB can cause lower respiratory tract damage in various ways, including direct injury via the TB pathogen, and indirectly via the host immune response [2,3]. Consequently, it can cause long-term respiratory problems. Recently, international awareness of PTLD has increased; however, much remains unclear [4]. Chronic pulmonary aspergillosis (CPA) is a major cause of PTLD.

CPA is an important pulmonary infection; it is estimated to affect up to three million people worldwide [5,6]. CPA is an emerging global concern because it is slow but progressive and it has a high mortality rate [7]. Prior PTB infection is the most potent risk factor for the development of CPA; there are an estimated 1.2 million people experiencing related sequelae [8,9]. CPA can also develop and become complicated after successful TB treatment [10,11,12].

CPA can be diagnosed using criteria based on a combination of multiple clinical characteristics [5,8,13]. Among these, anti-*Aspergillus* IgG antibody is critical because other microbiological studies, such as culture or biopsy, have low yield and cannot provide direct evidence of *Aspergillus* infection [13,14]. Moreover, it is considered a cornerstone of diagnosis because it can differentiate true CPA from colonized cases with high accuracy [15,16].

Previous studies evaluating the development of CPA described PTB, of varying rates, as an underlying respiratory condition, and these studies reported some demographic or radiographic features as risk factors [9,10,17,18,19]. However, these studies did not evaluate serological change in anti-*Aspergillus* IgG antibody in patients with PTB and its clinical usefulness in the diagnosis of CPA. For these reasons, in this study, we aimed to evaluate serological changes in anti-*Aspergillus* IgG antibody, before and after treatment of PTB, and the factors associated with the development of CPA after treatment completion.

## 2. Materials and Methods

### 2.1. Study Population

This study was a retrospective analysis of protocol-based collective data extracted from the TB clinic of Hallym University Kangdong Sacred Heart Hospital (Seoul, South Korea) from the period spanning January 2015 to December 2018. The hospital is situated in an area with an intermediate TB burden, as it has an estimated incidence of 77/100,000 persons in 2016 [20].

We enrolled consecutive patients aged (>18 years) who had been diagnosed with culture-positive PTB during the study period. In our TB clinic, all of the patients with PTB received anti-TB medication daily, and they underwent serum *Aspergillus* IgG investigation at least twice, before and after TB treatment. After treatment completion, the patients were followed up for at least 2 years to detect TB recurrence. During the follow-up period, they were interviewed about their symptoms and examined using chest radiography every 6 months. Patients with post-PTB complications or co-existing respiratory diseases, such as COPD, bronchiectasis, and advanced lung or pleural sequelae, were followed up and managed for more than 2 years, spanning beyond the 2-year follow-up period.

Patients with PTB were excluded if they were transferred from or out of another hospital, died or were lost to follow-up during anti-TB treatment, showed poor adherence to anti-TB treatment, had incomplete medical records, or if they had concomitant PTB and CPA at the time of PTB diagnosis.

The protocol for this study was approved by the Institutional Review Board (IRB) of Hallym University Kangdong Sacred Heart Hospital (IRB no. 2021–11-006). Patient information was anonymized and de-identified prior to analysis, and the requirement for informed consent was waived. Clinical, radiological, and laboratory data were collected anonymously using standardized forms, and follow-up data was obtained on 31 December 2020.

### 2.2. Measurement of Serum Anti-Aspergillus IgG Antibody Levels and Classification of Comparison Groups

Anti-*Aspergillus* IgG antibody was measured using *Aspergillus* precipitin tests (*Aspergillus fumigatus* immunoglobulin G enzyme-linked immunosorbent assay kit; IBL International, Hamburg, Germany). A positive result was reported as >12 IU/mL, whereas a negative result was reported as <8 IU/mL. Equivocal results were reported as 8–12 IU/mL, according to the manufacturer’s protocol. Results where the anti-*Aspergillus* IgG antibody level was within the level defined as positive were considered positive results, whereas results where the anti-*Aspergillus* IgG antibody level was within the level defined as equivocal or negative were considered negative results.

The enrolled patients were divided into four groups, based on the results of their serum anti-*Aspergillus* IgG antibody at baseline and at follow-up at the time of treatment completion, as follows: negative to negative, negative to positive, positive to negative, and positive to positive. Thus, the persistent seronegative patients were defined as those who were negative throughout; the seroconversion patients were defined as those who went from negative to positive; the persistently seropositive patients were defined as those who were positive throughout, and the seroreversion patients were defined as those who went from positive to negative (Figure 1).

### 2.3. Radiographic and Microbiological Evaluation

Chest radiographs and computerized tomography (CT) results of the patients were reviewed for the presence of cavities, and to determine the extent of the disease. If patients with PTB had a definite cavity on chest radiography, then a chest CT was not performed. In contrast, if it was unclear whether the patient with PTB had a cavity, then a chest CT was performed for confirmation. The extent of each lesion was categorized based on lobar involvement. Unilobar and multilobar PTB were defined as the involvement of ≤1 and ≥2 lobes, respectively. Acid-fast bacilli (AFB) smears were examined after auramine-rhodamine fluorescent staining, and they were graded on a scale from 0 to 4+. Mycobacterium tuberculosis culture was simultaneously performed using solid media, 3% Ogawa medium (Eiken Chemical, Tokyo, Japan), and liquid media in the mycobacteria growth indicator tube 960 system (BD Biosciences, Franklin Lakes, NJ, USA).

### 2.4. Diagnosis of CPA after PTB

The diagnosis of CPA was established based on the following criteria: (1) the presence of compatible clinical symptoms; (2) serological or microbiological evidence of Aspergillus infection, including positive serum anti-*Aspergillus* IgG antibody, isolation of *Aspergillus* species from respiratory specimens, or histological confirmation; (3) radiological findings compatible with evidence of disease progression; and (4) exclusion of alternative diagnoses according to the widely accepted diagnostic criteria proposed by the European Society for Clinical Microbiology and Infectious Diseases/European Respiratory Society [5,8,13].

### 2.5. Statistical Analysis

Data are presented as the median and interquartile range (IQR) for continuous variables, and as numbers (percentages) for categorical variables. Data were compared using the Mann–Whitney U test for continuous variables, and Pearson’s chi-square test or Fisher’s exact test for categorical variables. The Kaplan–Meier method and log-rank test were used to estimate the cumulative development rates of CPA, and to compare the rates between each serological group. Univariate and multivariate Cox proportional hazards regression analyses were performed in order to identify predisposing factors that were significantly associated with the development of CPA. Variables that were significant (*p* < 0.05) in the univariate analysis, and those of known clinical importance, were used in the multivariate analysis. The results are reported as hazard ratios (HRs) with 95% confidence intervals (CIs). All tests were two sided, and a P-value < 0.05 was considered statistically significant. Data were analyzed using IBM SPSS Statistics version 25 (IBM, Armonk, NY, USA), open-source statistical software Jamovi (version 2.2.2), and Graph Pad Prism 9.0 (Graph Pad, San Diego, CA, USA).

## 3. Results

### 3.1. Study Population and Baseline Characteristics

During the study period, 429 patients were diagnosed with, and treated for, culture-positive PTB, and, subsequently, 84 patients were excluded in accordance with the exclusion criteria. A total of 345 patients with PTB were included for analysis (Figure 1). Among the included patients, two patients underwent surgery owing to bronchopleural fistula and massive hemoptysis. The characteristics of the 345 enrolled patients are shown in Table 1. The median age of the patients was 55.0 years (IQR 37.0–69.0 years); 66.7% were male, and 56.5% were ever-smokers. The median body mass index (BMI) was 21.6 kg/m^2^. A history of previous TB infection was reported in 13.9% of the patients. The most common underlying disease was diabetes (20.9%), followed by chronic obstructive pulmonary disease or bronchial asthma (13.3%), and chronic liver disease (7.0%); among these, 37.4% and 48.1% were AFB smear-positive and had multilobar PTB, respectively. Of the patients with PTB, 27.8% had a cavity before treatment, and 11.9% still had a cavity after treatment. Chest CT was conducted in 191/345 (55.4%) patients at the time of diagnosis, and 156/191 (45.2%) patients underwent follow-up CT at the time of treatment completion.

### 3.2. Serological Changes before and after Treatment of PTB

A total of 256 (74.2%) patients with PTB tested negative for anti-*Aspergillus* IgG antibodies at the time of diagnosis, whereas the remaining 89 (25.8%) tested positive (Figure 1 and Figure 2). After treatment completion, anti-*Aspergillus* IgG antibody was measured for each patient during follow-up, and the overall seropositivity rate was 28.1%. In more detail, 9.4% (24/256) of patients who were seronegative for anti-*Aspergillus* IgG antibody at baseline were seropositive in the follow-up study, whereas 17.9% (16/89) of patients who were seropositive at baseline were seronegative in the follow-up study. Thus, the final serotypes were as follows: “persistently seronegative” in 232 patients (67.2%); “seroconversion” in 24 patients (7.0%); “seroreversion” in 16 patients (4.6%); and “persistently seropositive” in 73 patients (21.2%). To provide further insight into the longitudinal changes in the anti-*Aspergillus* IgG antibody, Figure 2b shows changes in anti-*Aspergillus* IgG antibody levels according to serological group.

### 3.3. Comparison of Clinical Characteristics between Each Serological Group

Table 2 shows the clinical characteristics of PTB patients according to their serological features (i.e., persistently negative, seroconversion, seroreversion, and persistently seropositive). As points of comparison among the four groups, the seroconversion and seroreversion groups had younger patients, and AFB smear positivity was more prevalent in the seroconversion group than in the other groups. Regarding radiological features, both cavity before treatment and multilobar involvement were more prevalent in the seroconversion group, whereas cavity after treatment was more prevalent in the seroconversion and persistently seropositive groups. Interestingly, the resolution rate of the primary cavity after TB treatment was 75.9% (44/58) and 40.0% (6/15) in the persistently seronegative and seroconversion groups, respectively; however, it was 0% and 22.7% in the seroreversion and persistently seropositive groups, respectively. Patients with previously treated TB were more prevalent in the persistently seropositive group than in the other groups. Never smokers were less common in the seroconversion group.

Table 3 shows a comparison of seronegative and seropositive groups based solely on the last result of the anti-*Aspergillus* IgG antibody. It shows that the seropositive group contained younger patients; this group was also more likely to have a cavity before and after treatment, and its members were more likely to have a history of previous TB infection.

### 3.4. Follow-Up and Development of CPA according to Serotype

The median (IQR) follow-up period after PTB treatment completion for all 345 patients was 25.8 (23.8–28.4) months. Regarding follow-up per group, it was as follows: 25.4 (23.0–29.0) months in the persistently seronegative group; 25.9 (24.3–33.4) months in the seroconversion group; 26.1 (24.1–27.9) months in the seroreversion group; and 25.8 (24.9–27.2) months in the persistently seronegative group. During the follow-up period, CPA developed in 10 patients (3%) (median 13.5 [3.0–27.3] months) after treatment completion. According to serotype, CPA developed in seven (7/24, 29.2%) cases in the seroconversion group, and it developed in three (3/73, 4.1%) cases in the persistently seropositive group (Figure 3). All 10 cases were positive for the *Aspergillus* precipitin test, two cases yielded positive cultures for *Aspergillus fumigatus*, and one case was confirmed following lung biopsy. CPA did not develop in the persistently seronegative and seroreversion groups during the study period. The cumulative development rate was higher in the seroconversion group than in the other groups (log-rank test, *p* < 0.001, Figure 4. Four cases of PTB recurrence were confirmed based on culture positivity during the follow-up period, and, from these, two (0.9%) were in the persistently seronegative group and two (8.3%) were in the seroconversion group).

### 3.5. Risk Factors for Development of CPA after PTB

The predisposing factors associated with the development of CPA were also explored (Table 4). In the univariate analysis, the factors that appeared to be significantly associated with an elevated risk of CPA after PTB included: high bacterial burden reflected in AFB smear positivity; seroconversion of anti-*Aspergillus* IgG antibody; chronic liver disease; and diabetes. In the multivariate analysis using the Cox regression model, both seroconversion of anti-*Aspergillus* IgG antibody (adjusted HR, 25.21; 95% CI, 6.11–103.99; *p* < 0.001) and diabetes (adjusted HR, 7.54; 95% CI, 1.93–29.50; *p* = 0.004) were significantly associated with the development of CPA after PTB.

## 4. Discussion

In this study, we aimed to evaluate serological changes in anti-*Aspergillus* IgG antibody in patients with PTB before and after treatment, and, additionally, we aimed to evaluate the risk factors associated with the development of CPA after treatment completion. We found that the baseline anti-*Aspergillus* IgG antibody results were negative in 74.2% and positive in 25.8% of patients with PTB. The serotype changed in 11.6% of patients (negative to positive in 7.0% and positive to negative in 4.6%) during anti-TB treatment. During follow-up, 10 patients (2.9%) developed CPA after PTB (Figure 5). The cumulative rate of CPA in patients with PTB was significantly influenced by the seroconversion of anti-*Aspergillus* IgG antibodies, and by diabetes.

As the burden of chronic respiratory disease, the primary underlying disease of CPA, increases worldwide, CPA is expected to become more common and receive more research attention [14]. However, despite CPA being a severe fungal infection, and a major cause of mortality among TB survivors, the precise prevalence of CPA after PTB is not well known, and this is due to its low incidence rate and the slow duration it takes to occur [6,10,21,22]. Thus, it is necessary to establish a large cohort with long-term observations to identify CPA, although this can be difficult in the real world [6,10]. Therefore, the disease burden of CPA can be estimated from significant studies previously undertaken in several countries [9,10,18,19,23,24,25]. There are various results regarding the prevalence of, and primary underlying diseases (such as PTB) in, CPA development. Many of the studies that were conducted in countries with a high TB burden were not observational studies, as they only reported estimates, and the largest study was performed 50 years ago in the UK; it spanned the period from 1960 to 1970. Therefore, there are still a lack of studies evaluating the prevalence of CPA which are also focused on post-PTB complications. Recently, a time-interval survey study performed in Uganda reported that CPA is a complication in 4.9% of patients treated for PTB; the prevalence rises to 26% in those with residual cavitation [10]. Another study, conducted in Indonesia, reported that 8% of patients had CPA at the completion of PTB treatment. This same study also provided evidence of serological changes in anti-Aspergillus IgG from baseline to treatment completion [26].

To better understand the precise disease burden of CPA, it is important to understand each element of its diagnostic criteria. CPA diagnosis will be performed via a combination of clinical characteristics until new innovative diagnostic methods emerge. Among these elements of diagnostic criteria, we focused on serological changes in anti-*Aspergillus* IgG antibody, the cornerstone of CPA diagnosis, displaying a positivity rate of over 90% in cases of CPA [5,17,27]. Prior studies have reported that the levels and results of anti-*Aspergillus* IgG antibody can change during or after TB treatment [10,28]. Another study demonstrated that patients who develop CPA can also become infected during TB treatment [29]. Based on prior studies, we believe that patients with PTB may be exposed to *Aspergillus* species during anti-PTB treatment following bronchial and parenchymal injury; therefore, further subclinical infection can occur and progress. If our hypothesis is reasonable, seroconversion of anti-*Aspergillus* IgG antibody, that is, serological change from negative to positive, may suggest a more aggressive lung injury, caused by TB, which is sufficient to cause exposure to, and recent infection with, *Aspergillus* species. Thus, we believe that it may be a new biomarker for CPA development. In our multivariate analysis, seroconversion of anti-*Aspergillus* IgG antibodies was significantly associated with the development of CPA after PTB.

In our analysis, residual cavity after PTB treatment did not show similar phenomena as in previous studies that described it as an important risk factor for CPA development [10]. This may be due to the relatively even proportion of patients with residual cavity in each group, and to the more defined residual cavities resulting from the large number of chest CTs. In our study, chest CT was conducted in 55.4% (191/345) of patients before treatment, and in 45.2% (156/345, including 87%, 156/191, of patients who underwent CT before treatment) of patients after treatment. As with the residual cavities in our study, the seropositive state of anti-*Aspergillus* IgG antibody after treatment did not show a significant association with the development of CPA after PTB. Considering these phenomena, we believe that *Aspergillus*-specific IgG positivity and characteristic CT findings, such as cavitation, are necessary and sufficient conditions for CPA diagnosis; thus, there may be no statistical differences and associations in our small sample size. Further studies are needed to clarify these disagreements and potential confounding factors.

There are some limitations to our study. First, because it was a retrospective hospital-based cohort study, and not a population-based study conducted at a single center, selection bias may have influenced our findings, and our results may not be generalizable to other countries. Second, this study was conducted in an area with an intermediate TB infection rate and a very low HIV infection burden. There were no cases of HIV infection in this study; this may have influenced our findings. Third, the observation period after completing treatment for TB may have been insufficient to ascertain the exact occurrence of CPA, considering the various occurrence periods of CPA [10,30]. Fourth, we used only one type of anti-*Aspergillus*-specific IgG antibody. Despite the significance of anti-*Aspergillus* IgG antibody detection using enzyme-linked immunosorbent assay (ELISA), and its strong recommendation by guidelines, there is no current definitive conclusion on each commercial test because of the difficulty in performing comparisons [5,8,27]. However, in South Korea, the commercial ELISA for *Aspergillus*-specific IgG antibodies produced by IBL (Germany) is widely used in most hospitals [31,32,33]. Fifth, there were only two cases of culture-positive *Aspergillus* species in our study. When we diagnosed CPA in patients treated for PTB, we attempted sputum culture for *Aspergillus* species, and bronchoscopy, if needed. However, the work-up was substandard because this study was retrospective in nature. This might have affected our results. Sixth, we only evaluated clinical information to predict the development of CPA; we did not perform a comprehensive assessment with laboratory science. This may be insufficient to explain our findings. Therefore, further translational prospective research is necessary in order to define the development of CPA and its predisposing factors, and to generalize our findings to other clinical situations.

## 5. Conclusions

The development of CPA in patients with PTB was observed in 2.9% of patients during follow-up after completion of treatment, and this development was significantly associated with both the seroconversion of anti-*Aspergillus* IgG antibody and diabetes.

## Figures and Tables

**Figure 1 jof-08-00130-f001:**
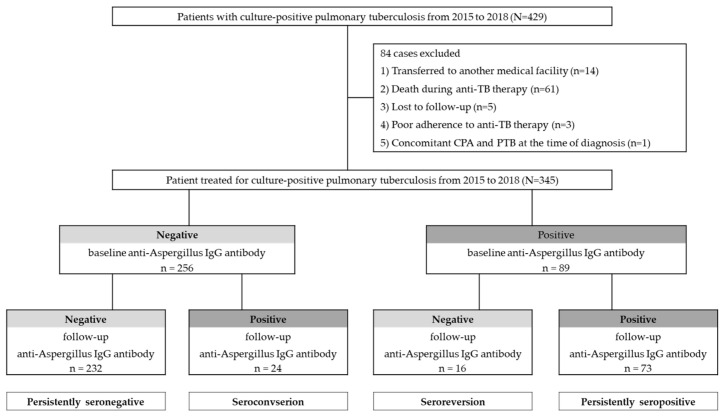
The study population according to results of anti-*Aspergillus* IgG antibody investigations.

**Figure 2 jof-08-00130-f002:**
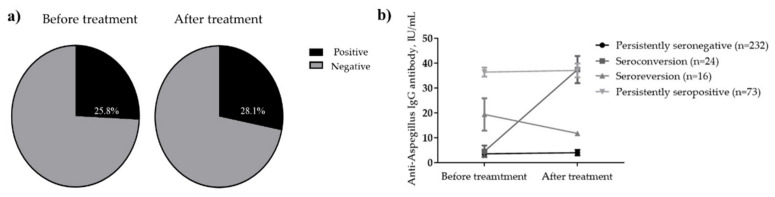
Seropositivity rates for anti-*Aspergillus* IgG antibody. (**a**) Change in levels according to serological group (**b**) before and after anti-TB treatment. In Figure 2b, anti-Aspergillus antibody levels are presented according to serotype as the mean with standard error of the mean.

**Figure 3 jof-08-00130-f003:**
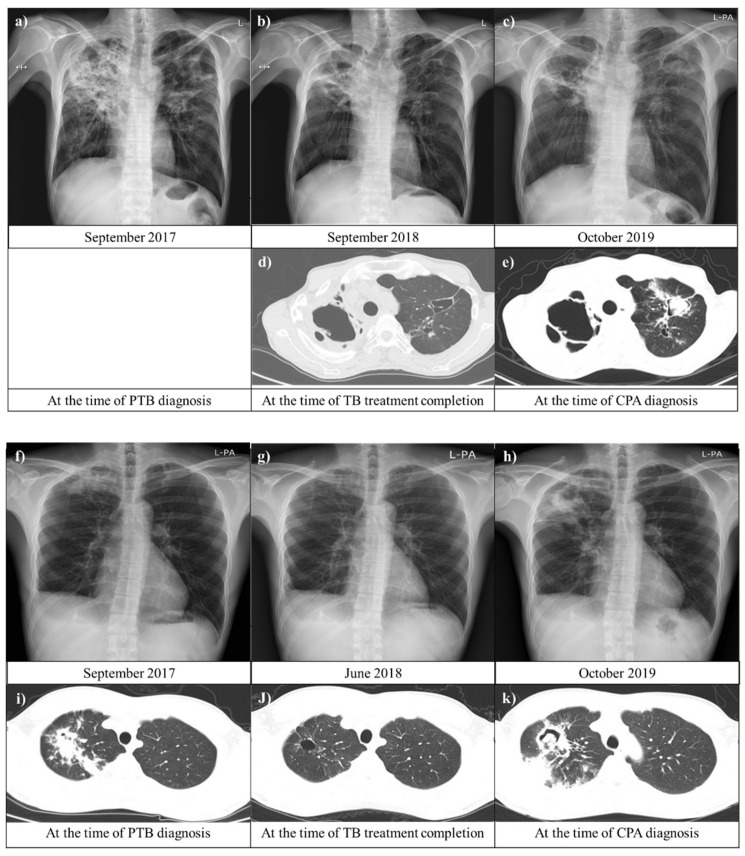
Chest images of patients with chronic pulmonary aspergillosis (CPA) after pulmonary tuberculosis (PTB). (**a**–**e**) A 53-year-old man was diagnosed with CPA after PTB in the seroconversion group. A chest image from September 2018 showed residual lung damage in both upper lungs. Approximately 11 months later, the patient had respiratory symptoms, including cough and productive sputum with hemoptysis. (**c**,**e**) display a new possible fungal infiltration in the scar in the left upper lung. (**f**–**k**) A 33-year-old man who was previously treated for PTB (five years prior) was diagnosed with CPA after a second instance of PTB in the persistently seropositive group. Serial chest images showed that primary PTB lesions resolved with anti-PTB treatment and aggravated with new fungal infection. Abbreviations: CPA, chronic pulmonary aspergillosis; and PTB, pulmonary tuberculosis.

**Figure 4 jof-08-00130-f004:**
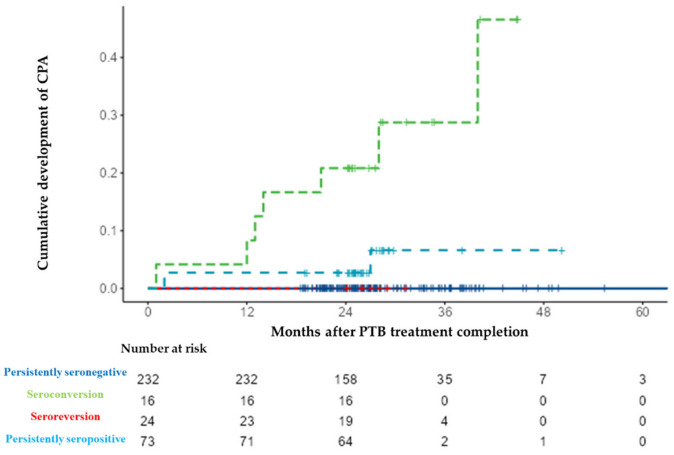
Cumulative development rate of chronic pulmonary aspergillosis (CPA) according to serological results for anti-Aspergillus IgG antibody. Abbreviations: CPA, chronic pulmonary aspergillosis; and PTB, pulmonary tuberculosis.

**Figure 5 jof-08-00130-f005:**
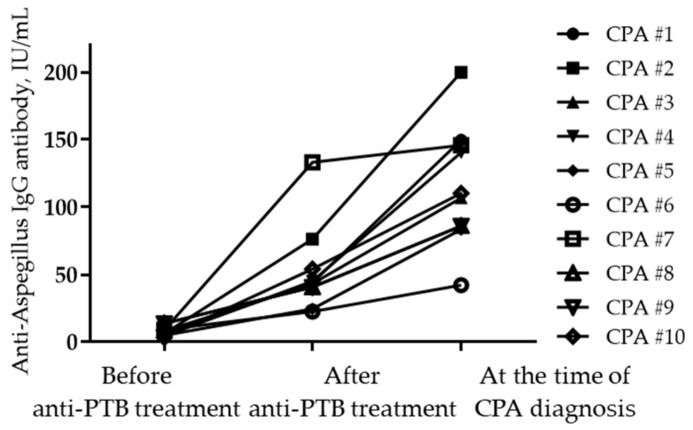
Changes in anti-*Aspergillus* IgG levels at different time points in patients diagnosed with CPA. Abbreviations: CPA, chronic pulmonary aspergillosis; and PTB, pulmonary tuberculosis.

**Table 1 jof-08-00130-t001:** Baseline characteristics of enrolled patients with PTB.

Characteristics	All Patients (*n* = 345)
Age, years Male sex, % Height, cm Body weight, kg BMI, kg/m^2^	55.0 (37.0–69.0) 230 (66.7) 166.0 (160.0–171.9) 59.0 (52.3–66.0) 21.6 (19.6–23.8)
Comorbidity	
COPD or asthma Thyroid disease Cardiovascular disease Malignancy Hematologic disease Chronic liver disease Rheumatic disease CKD Diabetes Neurologic disease Cerebrovascular disease KTP	46 (13.3) 3 (0.9) 20 (5.8) 12 (3.5) 4 (1.2) 24 (7.0) 13 (3.8) 15 (4.3) 72 (20.9) 10 (2.9) 12 (3.5) 2 (0.6)
Combined TB at another site	
PTB only PTB with EPTB	302 (87.5) 43 (12.5)
Smoking status	
Never smoker Former smoker Current smoker Cessation of smoking after diagnosis of PTB	150 (43.5) 72 (20.9) 123 (35.7) 71/123 (57.7)
Alcohol use (heavy alcoholics)	21 (9.5)
New cases Previously treated cases	297 (86.1) 48 (13.9)
Radiographic features	
Cavity (before treatment) Remaining cavity (after treatment) Extent of lung lesion, multilobar involvement	96 (27.8) 41 (11.9) 166 (48.1)
AFB smear status, positive	129 (37.4)
DST profiles	
Resistant to INH Resistant to RIF Resistant to INH and RIF	21 (6.1) 5 (1.4) 2 (0.6)
Treatment duration, months	6.5 (6.1–9.3)

Data are presented as median (interquartile range) or no. (%). Abbreviations: PTB, pulmonary tuberculosis; EPTB, extra-pulmonary tuberculosis; BMI, body mass index; AFB, acid-fast bacilli; COPD, chronic obstructive lung disease; CKD, chronic kidney disease; KTP, kidney transplantation; DST, drug sensitivity test; INH, isoniazid; and RIF, rifampicin.

**Table 2 jof-08-00130-t002:** Comparison of characteristics of patients with PTB according to change in serotype.

	Persistently Seronegative (*n* = 232)	Seroconversion (*n* = 24)	Seroreversion (*n* = 24)	Persistently Seropositive (*n* = 73)	*p*-Value
Age, years Male sex, % Height, cm Body weight, kg BMI, kg/m^2^,	57.0 (41.8–73.3) 153 (65.9) 166.0 (159.0–171.0) 58.1 (53.2–64.5) 21.6 (19.5–23.5)	46.0 (34.0–56.3) 19 (79.2) 166.0 (162.0–172.4) 55.4 (50.5–63.9) 20.3 (17.8–22.9)	43.5 (27.3–67.8) 11 (68.8) 164.0 (158.2.0–175.3) 59.0 (50.4–80.1) 22.5 (20.4–24.2)	50.0 (34.0–66.5) 47 (64.4) 167.1 (160.8–174.5) 62.1 (51.9–68.0) 22.0 (20.1–24.7)	0.003 0.609 0.383 0.118 0.087
before treatment					
BMI, kg/m^2^,	22.2 (20.6–24.5)	20.8 (19.7–23.5)	23.2 (20.7–24.4)	22.6 (20.4–24.7)	0.221
after treatment					
AFB smear status, positive	81 (34.9)	18 (75.0)	5 (31.3)	25 (34.2)	0.002
Radiographic feature					
Cavity (before treatment) Cavity (after treatment)	58 (25.0) 14 (6.0)	15 (62.5) 9 (37.5)	1 (6.3) 1 (6.3)	22 (30.1) 17 (23.3)	<0.001 <0.001
Extent of lung lesion,					
Multilobar involvement	110 (47.4)	20 (83.3)	7 (43.8)	29 (39.7)	0.002
New cases Previously treated cases	206 (88.8) 26 (11.2)	23 (95.8) 1 (4.2)	15 (93.8) 1 (6.3)	53 (72.6) 20 (27.4)	<0.001 <0.001
Smoking status,					
ever smoker	128 (55.2)	18 (75.0)	6 (37.5)	43 (58.9)	0.111

Data are presented as median (interquartile range) or no. (%). Abbreviations: CPA, chronic pulmonary aspergillosis; PTB, pulmonary tuberculosis; BMI, body mass index; AFB, acid-fast bacilli; COPD, chronic obstructive lung disease; CKD, chronic kidney disease; and KTP, kidney transplantation.

**Table 3 jof-08-00130-t003:** Comparison of the characteristics of patients with PTB according to the results of anti-*Aspergillus* IgG antibody investigations at treatment completion.

	Seronegative	Seropositive	*p*-Value
	*n* = 248	*n* = 97	
Age, years	57.0 (39.8–73.0)	49.0 (34.0–62.0)	0.002
Male sex, %	164 (66.1)	66 (68.0)	0.800
Height, cm	166.0 (159.0–171.0)	167.0 (162.0–173.5)	0.082
Body weight, kg	58.3 (52.9–64.9)	60.3 (51.6–68.0)	0.289
BMI, kg/m^2^, before treatment	21.7 (19.6–23.7)	21.4 (19.5–23.9)	0.788
BMI, kg/m^2^, after treatment	22.2 (20.6–24.5)	22.4 (20.0–24.7)	0.536
AFB smear status, positive	86 (34.7)	43 (44.3)	0.108
Radiographic feature			
Cavity (before treatment)	59 (23.8)	37 (38.1)	0.011
Cavity (after treatment)	15 (6.0)	26 (26.8)	<0.001
Extent of lung lesion, multilobar involvement	117 (47.2)	49 (50.5)	0.632
New cases	221 (89.1)	76 (78.4)	0.015
Previously treated cases	27 (10.9)	21 (21.6)	0.009
Smoking status, ever smoker	134 (54.0)	61 (62.9)	0.085

Data are presented as median (interquartile range) or no. (%). Abbreviations: CPA, chronic pulmonary aspergillosis; PTB, pulmonary tuberculosis; BMI, body mass index; AFB, acid-fast bacilli; COPD, chronic obstructive lung disease; CKD, chronic kidney disease; and KTP, kidney transplantation.

**Table 4 jof-08-00130-t004:** Analysis of predisposing factors related to the development of CPA after PTB treatment.

Variable	Univariate		Multivariate	
	HR (95% CI)	*p*-Value	HR (95% CI)	*p*-Value
Age, years	1.03 (0.97–1.39)	0.371		
BMI, kg/m^2^	1.16 (0.97–1.39)	0.107		
AFB smear status, positive	6.25 (1.32–29.55)	0.021		
Cavity (before treatment)	65,772,095.37 (0.35–infinity)	0.999		
Cavity (after treatment)	731,976,139.68 (0.00–infinity)	0.998		
Extent of PTB, multilobar involvement	2.23 (0.57–8.71)	0.249		
Seroconversion	27.97 (7.11–109.86)	<0.001	25.21 (6.11–103.99)	<0.001
Seropositive in treatment completion	324.70 (0.342–308,700.74)	0.098		
Previously treated cases	1.38 (0.29–6.54)	0.862		
Chronic liver disease	9.05 (2.54–32.19)	0.003		
Diabetes	5.81 (1.64–20.61)	0.007	7.54 (1.93–29.50)	0.004

Abbreviations: CPA, chronic pulmonary aspergillosis; PTB, pulmonary tuberculosis; and BMI, body mass index.

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
