# Peer review of "Serological Changes in Anti-Aspergillus IgG Antibody and Development of Chronic Pulmonary Aspergillosis in Patients Treated for Pulmonary Tuberculosis"

_jof, 2022, doi:10.3390/jof8020130_

Round 1
Reviewer 1 Report
The authors have followed over 340 culture proven TB patients to assess Aspergillus IgG seroconversion and development of CPA. They found 2.9% developed CPA just over a year after completing anti-TB therapy. As expected, this was linked to Asp. IgG seroconversion, but also diabetes. The % of patients with cavitation after TB was low at 11%.
Major comments
- The overall rate of CPA development was relatively low, but we don’t know how many patients died, had surgical resection of their TB or were lost to follow up. These data are required.
- L248-50. There has been recently published study of seroconversion during TB from Indonesia (Setianingrum et al, 2021).
- There is no commentary on the sensitivity and specificity of the Aspergillus IgG antibody test used (IBL). The manufacturer quotes an 81% sensitivity: https://www.ibl-america.com/content/elisa/IB79205.pdf and 91% https://www.ibl-international.com/media/mageworx/downloads/attachment/file/3/0/30113471_ifu_eu_en_aspergillus_fumigantus_igm_elisa_vn201707_sym4.pdf. The authors’ data might indicate a poor specificity with substantial numbers of persistently positive IgG cases, without clinical disease. The likely limitations of the chosen serology assay, and Aspergillus IgG in general should be discussed and addressed in future work, if the antibody performance is not clear (and not clear in Korea).
- An additional limitation may have been the lack of fungal culture of sputum (and antigen testing on bronchoscopy fluid) to diagnose possible CPA cases.
Some improvement in English required – examples:
L39/40
L75 -78 (were excluded comes twice)
L210/213/253 (use 10 not Ten)
Minor comments.
Figure 1 should come after Table 1.
Figure 2 legend is not clear for 2C. What do the colours represent? What the numbers in the boxes? Percentage of patients in each category?
Table 2 – some numbers appear to patient numbers with % and others actual IgG antibody levels. This is not clear. Also the bottom half of the table comparing all the comorbidities and the 4 serological categories is not necessary – simply add to the text.
Table 3 –same comment about co-morbidities.
Fig 4 – the axis labels are very hard to read. Perhaps Fig 4 should be split into 2 figures.
Author Response
Responses to Comments
Detailed description of the revision in accordance with the comments raised by the reviewer #1
The authors have followed over 340 culture proven TB patients to assess Aspergillus IgG seroconversion and development of CPA. They found 2.9% developed CPA just over a year after completing anti-TB therapy. As expected, this was linked to Asp. IgG seroconversion, but also diabetes. The % of patients with cavitation after TB was low at 11%.
Major comments
C1. The overall rate of CPA development was relatively low, but we don’t know how many patients died, had surgical resection of their TB or were lost to follow up. These data are required.
R1. We understand the reviewer’s concern. We have revised the Results section and modified Figure 1 to include this information according to your comment (shown in red font in the manuscript; page 5, lines 159–162 and pages 3–4, lines 140–144).
During the study period, 429 patients were diagnosed with and treated for culture-positive PTB, and subsequently 84 patients were excluded in accordance with the exclusion criteria. A total of 345 patients with PTB were included for analysis (Figure 1). Among the finally enrolled patients, two patients underwent surgery owing to bronchopleural fistula and massive hemoptysis.
The characteristics of the 345 enrolled patients are shown in Table 1.
C2. L248-50. There has been recently published study of seroconversion during TB from Indonesia (Setianingrum et al, 2021).
R2. Thank you for your comment. We were not aware of this study (Setianingrum F, et al. Thorax 2021;0:1–8.) when we conducted our research and wrote the manuscript. We have now cited the article and revised our Discussion section as shown below (also highlighted in red font color in the manuscript; page 10, lines 278–283):
This study is the first to evaluate serological change in anti-Aspergillus IgG antibody in patients with PTB before and after treatment of PTB, and the risk factors associated with the development of CPA after treatment completion.
Recently, a time-interval survey study done in Uganda reported that CPA is a complication in 4.9% of patients treated for PTB, with the prevalence rising to 26% in those with residual cavitation [10]. Another study from Indonesia reported that 8% of patients had CPA at completion of PTB treatment. It also showed serological changes in anti-Aspergillus IgG from baseline to treatment completion [26].
C3. There is no commentary on the sensitivity and specificity of the Aspergillus IgG antibody test used (IBL). The manufacturer quotes an 81% sensitivity: https://www.ibl-america.com/content/elisa/IB79205.pdf and 91% https://www.ibl-international.com/media/mageworx/downloads/attachment/file/3/0/30113471_ifu_eu_en_aspergillus_fumigantus_igm_elisa_vn201707_sym4.pdf.
The authors’ data might indicate a poor specificity with substantial numbers of persistently positive IgG cases, without clinical disease. The likely limitations of the chosen serology assay, and Aspergillus IgG in general should be discussed and addressed in future work, if the antibody performance is not clear (and not clear in Korea).
R3. Thank you for your comment. We have revised the Discussion section accordingly, as shown below (also shown in red font color in the manuscript; page 11, lines 321–327):
Fourth, we used only one type of anti-Aspergillus-specific IgG antibody. Despite the significance of anti-Aspergillus IgG antibody detection using enzyme-linked immunosorbent assay (ELISA) and its strong recommendation by guidelines, there is no current definitive conclusion on each commercial test owing to difficulty in performing comparisons [5,8,27]. However, in South Korea, the commercial ELISA kit for Aspergillus-specific IgG antibodies produced by IBL (Germany) is widely used in most hospitals [31-33].
C4. An additional limitation may have been the lack of fungal culture of sputum (and antigen testing on bronchoscopy fluid) to diagnose possible CPA cases.
R4. We understand the reviewer’s concern. We have revised the Discussion section accordingly, as shown below (also shown in red font color in the manuscript in red font; page 11, lines 327–331):
Fifth, there were only two cases of culture-positive Aspergillus species in our study. When we diagnosed CPA in patients treated for PTB, we attempted sputum culture for Aspergillus species and bronchoscopy, if needed. However, the work-up was sub-standard because this study was retrospective in nature. This might have also affected our results.
C5. Some improvement in English required – examples:
L39/40, L75 -78 (were excluded comes twice), L210/213/253 (use 10 not Ten)
R5. Thank you for your comment. As you recommended, we have gotten our manuscript further edited for English language.
Minor comments.
C6. Figure 1 should come after Table 1.
R5. Thank you for your comment. We have repositioned Table 1 and Figure 1 accordingly
(page 5, lines 155–162).
C7. Figure 2 legend is not clear for 2C. What do the colours represent? What the numbers in the boxes? Percentage of patients in each category?
R7. We understand the reviewer’s concern. We have deleted Figure 2C (alluvial plot), as it does not help to represent our data.
C8. Table 2 – some numbers appear to patient numbers with % and others actual IgG antibody levels. This is not clear. Also the bottom half of the table comparing all the comorbidities and the 4 serological categories is not necessary – simply add to the text.
R8. Thank you for your comment. We have removed the unnecessary data from Table 2.
C9. Table 3 –same comment about co-morbidities.
R9. Thank you for your comment. We have removed the unnecessary data from Table 3.
C10. Fig 4 – the axis labels are very hard to read. Perhaps Fig 4 should be split into 2 figures.
R10. Thank you for your comment. We have split Figure 4 into Figures 4 and 5
(pages 8–9, lines 235–241).

Reviewer 2 Report
The authors present a clinical study from South Korea in which the incidence of chronic pulmonary aspergillosis is documented in patients with previously treated tuberculosis over approximately two years. The found that all 10 patients who subsequently developed CPA had either persistent seropositivity to Aspergillus or because seropositive, indicating that the combination of certain CXR or chest CT changes and seropositivity to aspergillus spp is highly suggestive of or formally diagnostic of CPA. This is a valuable study, but the manuscript could be improved as follows:
- Certain statements made are either not correct or cannot be stated with any confidence due to the lack of supporting literature and thus should be deleted or modified. For example, it is fair to mention and cite literature pertinent to dysregulated proteases underlying PTLD, but it further needs to be mentioned that PTLD is a complex process involving a great deal more than just proteases that lead to lung destruction. Additionally, it not justified to state, “… CPA is the least known post-infectious complication of PTLD” when in fact is the one of the best known and most extensively described complications of PTLD.
- Figure 2. Please use a sans serif font (e.g., Calibri, Arial) for figures, this is much easier to read. As part C is already in color, suggest making the entire figure in color. This is especially true for part B, the groups of which are difficult to distinguish due to overlap; making the lines different colors will help relieve this issue. The alluvial plot (part C) is not explained and cannot be interpreted without a much fuller explanation of what this graph is showing. For example, what do the numbers mean? There appear to be 5 groups, but what are the different groups?
- Table 3. The text indicates when describing this table that that the seropositive group is younger, is more likely to have a cavity before and after treatment, more likely to have a history of previous TB infection, more likely to be an ever-smoker, and more likely to have chronic liver disease. However, this should be modified because:
- A P value is not listed for previously treated cases, thus the statistical difference, if any, is unknown. This statistic should be added and if not significant, this statement should be removed.
- The ever smoker difference is not significant (P = 0.085) and should not be called out.
- Comorbid associations. The authors note duly that CPA was linked to antecedent diabetes, but Table 4 clearly shows that chronic liver disease is even more significantly linked to CPA after PTB treatment. However only the significant link to diabetes is mentioned in the abstract and elsewhere. The manuscript should be amended to more inclusively and consistently note the link between diabetes and chronic liver disease and CPA.
Author Response
First Revision of jof-1535267
Responses to Comments
Detailed description of the revision in accordance with the comments raised by the reviewer #2
The authors present a clinical study from South Korea in which the incidence of chronic pulmonary aspergillosis is documented in patients with previously treated tuberculosis over approximately two years. The found that all 10 patients who subsequently developed CPA had either persistent seropositivity to Aspergillus or because seropositive, indicating that the combination of certain CXR or chest CT changes and seropositivity to aspergillus spp is highly suggestive of or formally diagnostic of CPA. This is a valuable study, but the manuscript could be improved as follows:
C1. Certain statements made are either not correct or cannot be stated with any confidence due to the lack of supporting literature and thus should be deleted or modified. For example, it is fair to mention and cite literature pertinent to dysregulated proteases underlying PTLD, but it further needs to be mentioned that PTLD is a complex process involving a great deal more than just proteases that lead to lung destruction. Additionally, it not justified to state, “… CPA is the least known post-infectious complication of PTLD” when in fact is the one of the best known and most extensively described complications of PTLD.
R1. Thank you for your comment. We have revised the Introduction section as shown below (also shown in red font color in the manuscript; pages 1–2, lines 36–48):
Unfortunately, survivors of PTB may experience post-tuberculosis lung disease (PTLD). TB can cause lower respiratory tract damage in various ways, such as via direct injury by the TB pathogen or indirectly via the host immune response [2,3]. Consequently, it can cause long-term respiratory problems. Recently, international awareness of PTLD has increased, while much remains uncertain.[4] Chronic pulmonary aspergillosis (CPA) is the major one of PTLD.
CPA is an important pulmonary infection, estimated to affect up to three million people worldwide.[5,6] CPA is an emerging global concern because it is slow but progressive and has high mortality.[7] Prior PTB infection is considered the most potent risk factor for the development of CPA with an estimated 1.2 million experiencing related sequelae.[8,9] CPA can also be developed and complicated after successful TB treatment [10-12]. However, CPA is the least studied post-infectious complication among PTLDs [4,6].
C2. Figure 2. Please use a sans serif font (e.g., Calibri, Arial) for figures, this is much easier to read. As part C is already in color, suggest making the entire figure in color. This is especially true for part B, the groups of which are difficult to distinguish due to overlap; making the lines different colors will help relieve this issue. The alluvial plot (part C) is not explained and cannot be interpreted without a much fuller explanation of what this graph is showing. For example, what do the numbers mean? There appear to be 5 groups, but what are the different groups?
R2. We understand the reviewer’s concern. We have removed Figure 2C as per the recommendation you and Reviewer#1 (C7) made. We also believe that the alluvial plot does not help to represent our data. Furthermore, we have modified Figures 1 to 5 by using Palatino Linotype font.
C3. Table 3. The text indicates when describing this table that that the seropositive group is younger, is more likely to have a cavity before and after treatment, more likely to have a history of previous TB infection, more likely to be an ever-smoker, and more likely to have chronic liver disease. However, this should be modified because:
- A P value is not listed for previously treated cases, thus the statistical difference, if any, is unknown. This statistic should be added and if not significant, this statement should be removed.
- The ever smoker difference is not significant (P = 0.085) and should not be called out.
R3. We apologize for the missing information. We have included the P-values for variables of previously treated cases. We have removed the phrase “more likely to be an ever-smoker, and more likely to have chronic liver disease” because Reviewer#1 also suggested that the comparison of comorbidity is unnecessary in Tables 2 and 3 (page 6, lines 193-196 and pages 6–7, lines 201–205).
C4. Comorbid associations. The authors note duly that CPA was linked to antecedent diabetes, but Table 4 clearly shows that chronic liver disease is even more significantly linked to CPA after PTB treatment. However only the significant link to diabetes is mentioned in the abstract and elsewhere. The manuscript should be amended to more inclusively and consistently note the link between diabetes and chronic liver disease and CPA.
R4. We understand the reviewer’s concern. We found statistical differences in AFB smear status, seroconversion of anti-Aspergillus IgG antibody, chronic liver disease, and diabetes using univariate cox regression. However, the multivariate cox regression model showed that only seroconversion and diabetes were statistically different. This result, which shows that diabetes is a significant predisposing factor of CPA in patients with PTB, is similar to those in recently published reports from Indonesia (Setianingrum F, et al. Thorax 2021;0:1–8.). Please note the C2 of Reviewer#1 and R2.

Round 2
Reviewer 2 Report
Most concerns have been addressed through explanation or modification of the manuscript, but the authors still refer to CPA as the least studied (previously, "known") of the PTLD complications. Again, this is just not true-there are numerous descriptive studies of CPA following tuberculosis in relation to other PTLD complications. Dr. Denning, who the authors liberally cite, is just one of many scientists who have published on exactly this topic. The authors might wish for better and higher quality data on the topic, which is fair, but that does not mean the subject has not already been studied extensively. As a compromise, suggest just dropping this sentence all together-it will not compromise the manuscript in any way.
Author Response
Second Revision of jof-1535267
Responses to Comments
Detailed description of the revision in accordance with the comments raised by the reviewer #2
C1. Most concerns have been addressed through explanation or modification of the manuscript, but the authors still refer to CPA as the least studied (previously, "known") of the PTLD complications. Again, this is just not true-there are numerous descriptive studies of CPA following tuberculosis in relation to other PTLD complications. Dr. Denning, who the authors liberally cite, is just one of many scientists who have published on exactly this topic. The authors might wish for better and higher quality data on the topic, which is fair, but that does not mean the subject has not already been studied extensively. As a compromise, suggest just dropping this sentence all together-it will not compromise the manuscript in any way.
R1. Thank you for your detailed comment. We understand the reviewer’s concern and agree. We excluded that sentence in the manuscript.
